# Transreligiosity and the Messiness of Religious and Social Worlds: Towards a Deleuzian Methodological Imagination for Religious Studies

**Paul-François Tremlett**

Department of Religious Studies, Faculty of Arts and Social Sciences, School of Social Sciences and Global Studies, The Open University, Milton Keynes MK7 6AA, UK; paul-francois.tremlett@open.ac.uk

**Abstract:** Research methods and concepts in religious studies are conventionally understood as procedures and rules for representing religious and social worlds. However, religious and social worlds are simultaneously messy, lively and elusive, and arguably transreligious ones are especially so. In this essay I reflect on Panagiotopoulos and Roussou's concept of "transreligiosity" as a means for re-thinking classical and contemporary methodological debates in religious studies, and for reflecting on methods as social practices.

**Keywords:** transreligiosity; crisis; method; Deleuze; nomadology; ontological politics

## 1. Introduction

The objective of this essay is to explore the concept of "transreligiosity" in relation to methodology. If metaphors and allegories of elasticity, porosity and hybridity are appropriate for re-thinking religion at a time of multiple and interlinked political, economic and knowledge crises, might those same metaphors and allegories enable a productive re-assessment of how scholars of lived and vernacular religion conceptualise their research methods? Research methods and concepts in religious studies are conventionally understood to be procedures and protocols for representing religious and social worlds. However, Mol's contention that "reality does not precede the mundane practices in which we interact with it, but is rather shaped within these practices" (Mol 1999, p. 75), suggests an alternative point of departure in which methods are not simply implicated in the representation of or in bringing multiple perspectives to bear on a single underlying reality, but are rather implicated in the very enactment of realities. This essay begins by outlining some of the key features of transreligiosity before sketching a history of methodological debates in religious studies. It then turns to the work of Deleuze and Guattari and their conception of nomadology as a means of exploring the enactive, ontological politics of research methods as social practices and their consequences.

## 2. Transreligiosity and the Question of Method

What kinds of religious and social worlds does "transreligiosity" open and make visible? Panagiotopoulos and Roussou's provocation that "*we have never been religious*" assumes the impossibility of any singular or pure, religious object or form. Despite manifold efforts—in both secular and religious contexts—to centralise, institutionalise and purify the religiousness of religion, they point to a proliferation of phenomena at the vernacular level which they organise in terms of metaphors and allegories that evoke liquids, porosity, webs, flows, elasticity, transgression and hybridity. For Panagiotopoulos and Roussou, "transreligiosity" encompasses religion, spirituality, the New Age and lived religion, and reveals frictions between an "institutionalized, centralized, and dogmatic class of 'intermediaries' which defines what is religion, [and] what is not", and a "counter-domain full of 'hybrids', namely, of non-official, decentralized, non-dogmatic constellations" of

transreligion (2022, p. 4). Panagiotopoulos and Roussou's examples of transreligiosity include Afro-Cuban religiosity which they describe as "open", "idiosyncratic" and "multiple" (2022, p. 5), and New Age spirituality and alternative medicine in southern Europe, which they describe in terms of the transgression of "religio-spiritual boundaries" either "actual, via travelling across geographical frontiers" or "symbolic, through the creative amalgamation of different religious traditions" (2022, p. 11).

Panagiotopoulos and Roussou's centring of transreligiosity also points to emergent developments in theology. According to Kalsky (2017), in the Netherlands, interrelated processes of secularization, individualization and mobility have seen the emergence of new forms of religious identification based less on any vertically organised connection to a single religious institution, than flexible, horizontal connections to multiple sources of religious authority, knowledge and practice to form what is termed, "multiple religious belonging" (see Grant 2018 but also Nunes 2021). Kalsky cites survey research from 2015 estimating that 23 percent of the Dutch population—some three million people—"see themselves as combiners of elements from various religious traditions" (Kalsky 2017, p. 346), and she suggests that the prefix "trans" highlights "the flowing and flexible shape of hybrid religious identities" (2017, p. 357). Kalsky imagines transreligiosity as a "rhizomatic network" (2017, p. 356) of lines, drawn from the biographical choices of individual actors, evoking a sense of religious and social worlds less as fixed structures that align individuals with the concrete social facts of institutions and traditions, than an effervescent field of contingent connections made visible in the choices and actions of individuals.

Chris Cotter's compelling study of non-religion discloses yet another layer to this transreligious complexity: Cotter characterizes non-religious identities as "relational acts" (2020, p. 137) emergent in particular historico-spatial contexts as opposed to fixed or unitary substances (2020, p. 11), and non-religion as something performative, relational and situational (2020, pp. 204–5), as opposed to an invariant set of beliefs or dispositions. Non-religion, then, is less a thing or an object or a substance than a moment, a flow or a liquid, an elusive, emergent node that articulates and networks, a potential that may form a solid structure but may, by the same token, simply dissipate into airy nothingness. Afterall, there are no standard texts, buildings or public spaces that belong to or identify non-religion, and no defining beliefs or stable, self-identifying populations. Cotter's *vignettes* taken from his encounters and interviews in Edinburgh's Southside and the careful analysis of the words of his interlocutors, deftly opens out the improvised and provisional quasi-being of non-religion.

For Panagiotopoulos and Roussou, as well as for Kalsky and Cotter, religions, religious and non-religious identities—if not the entirety of the social itself—appear to have lost their solidity, such that what seemed once to be discrete sociological and theological objects have become hopelessly entangled and mixed up. If all that was solid has not exactly melted into air, the concept of transreligiosity would seem to point to a multiplicity of religio-social forms, where the traditional objects of religious studies have transmogrified and become liquid. However, amidst the, by turns, messy, lively and elusive religio-social worlds they each describe, none ask whether the rules and protocols of nineteenth and twentieth century research methodologies that have historically transcribed religio-social worlds as being populated by concrete identities, roles and institutions defined by fixed social practices and discrete beliefs, are the most appropriate ones for approaching the interlinked economic, political and knowledge crises of "liquid modernity" (Bauman 2000). As Robert Cooper has argued, our "institutional skills favour the fixed and static, the separate and self-contained. Taxonomies, hierarchies, systems and structure represent the instinctive vocabulary" of disciplinary thought, which constitutes "a world of finished subjects and objects from the flux and flow of unfinished, heteromorphic" life (Cooper in Law 2004, p. 104). Of course, perhaps there are sometimes good reasons to foreground the fixed, the solid and the invariant in religious studies, but the question about methods becomes more salient when they are understood as social practices that are not applied to a static or passive reality, but as ones that "interfere" (Mol 1999, p. 74) with and enact it.

### 3. Classical and Contemporary Methodological Debates in Religious Studies

In *After Method: Mess in Social Science Research* (2004), John Law reflects on what happens when social scientists try to describe or represent the social worlds which are "complex, diffuse and messy" (2004, p. 2). According to Law, current research methods are not well adapted to the study of "the ephemeral, the indefinite and the irregular" (2004, p. 4). Part of the reason for this is that "methods, their rules, and even more methods' practices, not only describe but also help to *produce* the reality that they understand" (2004, p. 5 italics in original; see also Law and Urry 2004; Coleman and Ringrose 2013). It is important to remember that the very inception of the sociological imagination by Auguste Comte was intended not as a means of describing a particular social situation, but of enacting a new one. Writing in the aftermath of the French Revolution, Comte sought to hasten the emergence of a new, rational and modern, industrial–scientific social order. The old, feudal formation of aristocracy, Church, and monarchy, with its arbitrary privileges, had been only partially eclipsed in the violent energies of the revolution of 1789. Comte saw an opportunity to bring an end to the uncertainties of the times by establishing a new, industrial society, built on rational–secular principles that would be led by, among others, sociologists. According to Comte, the new science of sociology was needed to reorganize society by raising "*politics to the rank of the sciences of observation*" (Comte 1998a, p. 81, italics in original). Initially he called it "social physics" (Comte 1998b, p. 158), but toward the end of the *Course in Positive Philosophy* (1830–1839) he coined the term "Sociology". Comte's sociology drew its initial methodological inspiration from physiology but by the late 1830s he had fleshed out a vision of the new science to include "statics" and "dynamics", the former to analyse the structure of a given society and the latter to consider its historical development. In short, the discipline of sociology and its formative methods did not emerge simply to describe modernity, but rather to perform it.

Two points follow from the above; the first is the idea that current research methods may be ill-equipped to study certain kinds of hyper-complex phenomena which could elude the grasp of certain kinds of research methods. The second is the idea that research methods do not enable so much the description or representation of states of affairs in the world, as the performance or enactment of them. Importantly, both of these points are anticipated in classical methodological debates in religious studies, which have conventionally been conducted in terms of Wilhelm Dilthey's classical methodological opposition of the *Naturwissenschaften* to the *Geisteswissenschaften*—the contrast of the natural sciences to the arts and humanities—and of explanation (*Erklärung*) to understanding (*Verstehen*) (Palmer 1969, p. 100). For Dilthey as for and others writing in the phenomenological and hermeneutic traditions, whereas the natural sciences focus on explaining laws of cause and effect between phenomena, the arts and humanities are concerned with understanding the special meanings and inner (spiritual) experiences of human beings that are held to lie behind all great art, literature and revelation (see Grondin 1994; Palmer 1969; Harrington 1996, p. 27). Accordingly, this kind of knowledge cannot be reduced to or be contained by the formal procedures of the sciences. Moreover, when it comes to the study of religion and the sacred, not only are special methods required to apprehend them but also literary techniques of affective evocation that exceed formal description or representation are necessary to, as it were, enact them. As such, Gold has suggested that phenomenologists of religion have exploited a certain friction between representation and evocation, in the process producing a distinct genre of writing that "plays on the tension between a romantic evocation of the human imagination and a rationally enlightened, scientifically true, analysis" (Gold 2003, p. 45). Through reference, then, to methodological principles such as bracketing, an aura of presuppositionless neutrality can be sustained even as the phenomenologist's own language is acknowledged as "performative":

> The phenomenologist's evocative description turns out to be a type of performative language intimately tied to his method of enquiry: the phenomenologist uses language in a quasi-causal way to evoke or prompt the reader's own empathetic response and appreciation of aspects of religious consciousness. This performa-

tive use of language to describe evocatively is indicative of the phenomenologist's respect for showing the phenomenon as it appears in religious consciousness as well as his methodological commitment to experiential understanding of the structures of human consciousness. (Twiss and Conser 1992, p. 13)[1]

An exemplar of this kind of approach can be found in the figure of Mircea Eliade, who was one of the most influential and remains one of the most controversial figures in the study of religions. His writings articulate a profound sense of spiritual crisis but, for Eliade, it is a crisis that can be averted if human beings can be re-connected to the deep truths from which they have become estranged in modernity. In order for this restoration to occur, however, it is imperative to resist the "interpretations of religious realities made by psychologists, sociologists, or devotees of various reductionist ideologies" (Eliade 1969a, p. 70). This is the heroic role Eliade foresaw for himself and the study of religions—to restore the "total man" to the Sacred:

> It seems to me difficult to believe that, living in a historical moment like ours, the historians of religions will not take account of the creative possibilities of their discipline. How to assimilate *culturally* the spiritual universes that Africa, Oceania, Southeast Asia, open to us? All these spiritual universes have a religious origin and structure. If one does not approach them in the perspective of the history of religions, they will disappear as spiritual universes; they will be reduced to *facts* about social organizations, economic regimes, epochs of precolonial and colonial history, etc. In other words, they will not be grasped as spiritual creations; they will not enrich Western and world culture—they will serve to augment the number, already terrifying, of documents classified in archives, awaiting electronic computers to take them in charge. (1969a, pp. 70–71; italics in original)

According to Eliade, religion is a unique and irreducible phenomenon that in a certain sense resists analysis, such that its core or essence will always elude capture by anything other than "a special hermeneutics" (Eliade 1969b) pre-attuned to the deep reality of the sacred. Arguing that the phenomenology of religion deals not with "fossils" or "ruins" but rather (religious) "messages" that "disclose fundamental existential situations that are directly relevant to modern man", he suggests that the work of interpreting these messages can occasion "the inner transformation of the researcher and, hopefully, of the sympathetic reader" such that this special hermeneutics should be understood as a "propaedeutic and spiritual" technique in its own right (Eliade 1969b).

However, during the 1990s and early 2000s a new kind of religious studies scholarship emerged that was highly critical of Eliade, phenomenology and hermeneutics (see Cho and Squier 2008). This new religious studies took two principal forms and directions. For both there was no special, elusive something that would always elude analysis, while the purpose of method was not to disclose any sacred reality but to more accurately represent the basis of religious beliefs and practices, be it located in the architecture of the mind or in the contingent constructions of discourse. The first, then, took a Kantian or cognitive turn to argue that religion was ultimately epiphenomenal of innate and invariant psychological structures, drawing substantial impetus from "religion scholars wanting to 'science up' the study of religion" (Barrett 2011, p. 230). According to Jensen, cognitive approaches are broadly in keeping with "classificatory phenomenologies of religion" (Jensen 2014, p. 78), although cognitive theory's "standard model" "posits religion as a by-product of other evolutionary adaptations" (Jong 2017, p. 54) and "religious thoughts" not as "a dramatic departure from, but a predictable by-product of, ordinary cognitive function" (Boyer 2003, p. 119), which is precisely the kind of reductionism Eliade and others in the phenomenological tradition were seeking to rule out.[2]

The second, known as discourse analysis, drew significantly from Michel Foucault's archaeology and indeed on traditions of textual analysis that recalled the study of religions' philological heritage, to investigate the historical formation of religion as a category and its implication in colonial and nationalist projects, to attend to the different ways in

which the term has been mobilised and put to work in contemporary contexts and also to bring "explicit and testable theories of religion" McCutcheon (1997, p. 193) to the table (see Cho and Squier 2008). Informed, then, by post-structuralist critiques of the subject and Marxist and Foucauldian critiques of power, discourse analysis took broadly two forms, on the one hand the analysis of "historical, large-scale discursive frameworks" and on the other, the "close reading of texts" (Taira 2016, p. 127) (for an excellent summary of the approach and an outline of its intellectual genealogies, see Hjelm 2016). McCutcheon's claim that

> the common assertion that religion per se or private religious experience in particular, is sui generis, unique, and sociohistorically autonomous, is itself a scholarly representation that operates within, and assists in maintaining, a very specific set of discursive practices along with the institutions in which these discourses are articulated and reproduced. (1997, p. 3)

constitutes a fair indication of the point of departure of the approach, which sees religion and the sacred as social constructions typically produced in moments of social contention. The centrality of ideology critique to much of this work is also notable (see Fitzgerald 2003; Martin 2022).[3]

The cognitive turn and discourse analysis, then, emerged to challenge the hegemony of phenomenology and hermeneutics by claiming a kind of legitimacy crisis for religious studies that could only be addressed through the adoption of allegedly more scientifically rigorous methods and approaches. However, what I am interested in here is the extent to which these different approaches—as social practices—enact religious and social worlds in specific ways. It is important to grasp that this does not mean that they—phenomenology, cognitivism and discourse analysis—approximate three different perspectives or points of view on the same object. Rather than imagining methods as distinct lenses that gaze upon a single, underlying reality which remains untouched throughout, we need to consider that "reality is manipulated by means of various tools in the course of a diversity of practices" (Mol 1999, p. 77), including methodological practices.

## 4. Nomadology and the Deleuzian Imagination

Phenomenology, hermeneutics, cognitive theory and discourse analysis (among others) are certainly legitimate methods for investigating religion, but they are also in denial about themselves and their methods as social practices, and the enactments they make. Religious and social worlds—like all worlds—emerge relationally or, in Law and Urry's words, "*reality is a relational effect*" (2004, p. 395; italics in original; see also Mol 1999; see also Rovelli 2021).[4] They go on to suggest that "once we start to imagine methods in this way we enter the realm of ontological politics" (2004, p. 396), a realm where the different conclusions generated by different research methods cease to point to different standpoints about a single reality and instead imply that "the world is multiply produced in diverse and contested social and material relations" (2004, p. 397). Importantly, these multiple worlds or realities do not co-exist side by side but are mutually implicated:

> States of things are neither unities or totalities, but *multiplicities*. It is not just that there are several states of things (each one of which would be yet another); nor that each state of things is itself multiple (which would simply be to indicate its resistance to unification). The essential thing, from the point of view of empiricism, is the noun *multiplicity*, which designates a set of lines or dimensions which are irreducible to each other. Every 'thing' is made up in this way. (Deleuze and Parnet in Coleman and Ringrose 2013, p. 9; italics in original)

This ontological and relational multiplicity is also a reflexivity that understands methodologically constituted relations as always already embedded in asymmetries and territorializations—issues well understood in post-colonial (Smith 1999) and feminist theory (Haraway 1991)—and which points to methods not as technical instruments or protocols, but as performances that may congeal existing asymmetries and territorializations or enact

new ones. Elsewhere I have argued for a "nomad science of religions" (Tremlett 2021, p. 151; Tremlett 2022; see Deleuze and Guattari 2014) that does not begin with a world out there that is, in theory, possible to delimit and define in its entirety. Rather, I suggested that nomadology begins with "interactions" (2021, p. 153) whose open-endedness precludes definitiveness and which points to complexities that cannot be reduced to linear notions of causality. Panagiotopoulos and Roussou's concept of transreligiosity resonates with nomadology to the extent that it presupposes religious and social worlds constituted through unpredictable forces and flows that generate new objects and combinations of objects to make hybrids (see also Law 2004, p. 137). Attentiveness to processes of composition and decomposition—that is, to processes of transreligious formation and assemblage—is a critical feature of nomadology (Ruddick 2012). Deleuze and Guattari (2014, pp. 420–36) distinguish between nomad science and royal or sovereign science, stating that while the latter deals with laws, axioms and legislation, the former attends to the disruptive flows and transgressions of hybrids. Therefore, what does a focus on composition/de-composition look like?

> Nomad science emphasizes the malleable, fluid and metamorphic nature of being, while state science conceptualizes being as solid, essential and unchanging. Nomadology is the study of wandering subjectivities, of beings that drift from predetermined or normative paths, particularly those paths determined and regulated by apparatuses of the state. For Deleuze and Guattari, nomadism is a form of life that is shaped by continual embarkation on lines of flight—that is, modes of escape, moments of transformation, ways of becoming other-than-normative and ways of acting in excess of, or insubordinately in relationship to, repressive forces. Lines of flight have the capacity to deterritorialize, to undo, to free up, to break out of a system or situation of control, fixity or repression. Nomad science, by extension, concerns itself with experiments and inventions that are fundamentally deterritorializing, while state science is, by counterpoint, fundamentally reterritorializing. To territorialize an entity is to set and define its limits, to organize component parts into a coherent whole determined by a specific end. (Malatino 2014, p. 138)

According to Cole, "nomadic analysis" (Cole 2013, p. 219) in ethnography entails a focus "on the flighty . . . sometimes miniscule comments, moments and asides" (Cole 2013, p. 235) that seem not to belong anywhere and which somehow evade capture on the analytic grid (I might add that it does not have to be words or gestures—it could equally be an item of material culture). Similarly, Law (2004) writes of a "method assemblage" (2004, p. 42) which he describes as a "radio receiver, a gong, an organ pipe, or a gravity wave detector, a set of relations for resonating with and amplifying chosen patterns which then return to the flux" (2004, p. 117), which enables the researcher to attend both to what her methods are drawing out, and to what they are silencing. Importantly, this compositional dimension also points to the ontological politics of multiplicity.

In the late 1990s I was funded by the British Academy to conduct field research on alternative, rural religiosities at an extinct volcano some seventy kilometres south of Manila, called Mount Banahaw. I spent a lot of time with healers as well as small, independent churches and spiritual groups such as the Ciudad Mistica de Dios (the Mystical City of God, see Quibuyen 1991), the latter notable for their millenarian conviction in the gendered transformation of society and their veneration of José Rizal—Filipino novelist, ophthalmologist and national hero who was executed by the Spanish colonial regime in 1896—at a site popularly associated with pilgrimage to its sacred shrines and for the accumulation (and distribution) of sacred power and potency (*kapangyarihan*), particularly during Easter.

This hybrid combination of Christian millenarianism, feminism, local conceptions of power, healing and nationalism with an urban imaginary, is an exemplary instance of transreligiosity. Previous scholarship on Mistica and groups like it at Banahaw and elsewhere has claimed them to be the result of a "complex clash between customary and



modern tendencies . . . by-products of the stress between . . . the little and great traditions" (Sturtevant 1976, p. 17), and a product of a potentially catastrophic cognitive divide between modern, elite forms of knowledge and experience and those of the "masses" that could only be understood and potentially healed by "bring[ing] to light the masses' own categories of meaning" (Ileto 2011, p. 8). It was taken for granted that the little tradition, like the categories of meaning of the masses, would inevitably succumb to the corrosive flows of modernity into which they would eventually dissolve and disappear. In short, a notable objective of ethnographic research about groups such as Mistica and sites like Banahaw was hardly innocent, oriented as it was to tabulating and documenting the uneven progress and even reversals of modernity, while enacting its ontological politics through the production of religion as a barometer of stress and irrational reflex.

On my journeys in and around the mountain I had many conversations with Mistica's Secretary General. On one particular occasion we discussed, over coffee and *merienda*, the eschatological fate of those Filipinos who had never heard the word of God because they had had the misfortune of having lived long before the arrival of Spanish missionaries. It was not unusual for us to explore, in our conversations, the historical experiences of the Philippines in relation to mission and colonialism. My interpretive framework postulated that Banahaw was a product of powerful, historical forces whose tides had swept Filipinos into a storm of change and upheaval. The mountain with its millenarian and magical reputations was a reaction with and against those forces (Tremlett 2002). Then, he said something that, at the time, I did not understand. He said, "salvation is a right". Those four words—as I understand them now—opened out an alternative. "Salvation is a right" made salvation independent of time and place disclosing a connected reality, one constituted not by the vagaries of violence, chance or reflex, but by justice. I am still reflecting on how I might have and might yet engage methods to make justice something real:

> If methods are not innocent then they are also political. They help to *make* realities. But the question is: which realities? Which do we want to help to make more real, and which less real? How do we want to interfere (because interfere we will, one way or another)? (Law and Urry 2004, p. 404)

My second example comes from much more recent online research conducted with groups and organisations involved in campaigning on human rights issues in the Philippines. I was interested in how these groups and organisations go about assembling or composing "moral publics" (Jedlowski 2018) through which to highlight human rights abuses and generate traction to have those abuses addressed, and the role of affect in those processes. They shared with me how they mobilised their resources to create sites and occasions around which publics might coalesce to try to secure particular social, developmental and political objectives. In one interview we were exploring some of the differences between digital and face-to-face repertoires of protest and campaigning, and one of the activists put it to me that physical protests and events "somehow takes your one voice and unites it with the other voices who are chanting the same thing and creates that, you know . . . common experience". On the one hand he articulated an activist metaphysics of presence for which online repertoires of protest were essentially derivative of face-to-face ones. But on the other, in just a few words, he reminded me that research is not simply the enumeration of facts, in this case, facts about protests, politics and movement building. Rather, by employing a vocabulary of experience and community, I was being tuned out of a certain kind of political rationality that privileged the subject and the individual choice-making actor and tuned into a connected reality, where politics afforded a certain possibility for this actor to be immersed, however temporarily, into a different kind of sensuously embodied, hearing subject.

These brief examples of "nomadic analysis" are important because they made felt resonances I had not anticipated and which my research methods had not really been disposed to hear. In both instances my attention was drawn to the compositional and de-compositional processes through which subjects and objects are formed—that is, both to the composition of the kinds of transreligious objects highlighted by Panagiotopoulos

and Roussou (2022) but also to that of different kinds of subject as presupposed by diverse methodological practices, and thereby to the enactive affordances and the ontological politics of method.

## 5. Conclusions

In this essay I have suggested that research methods and concepts in religious studies are conventionally understood as procedures and rules for representing religious and social worlds. However, religious and social worlds are complex, messy and elusive, and arguably transreligious ones are especially so. Reflecting on Panagiotopoulos and Roussou's (2022) concept of "transreligiosity" and classical and contemporary methodological debates in religious studies I have argued, following Mol (1999), Law (2004), Law and Urry (2004) and Deleuze and Guattari (2014), that research methods are enactive social practices. They are not protocols or rules to be applied to a pre-existing external reality but are the very means through which reality is made. The fact that Panagiotopoulos and Roussou organise "transreligiosity" in terms of metaphors and allegories that evoke liquids, porosity, webs, flows, elasticity, transgression and hybridity reflects the interlinked political, economic and knowledge crises that augur the demise of a more or less "nineteenth century imagination and metaphysics, which assumed that the world is just out there, more or less given, and it is the job of the scientist . . . to map reality" (Law and Urry 2004, p. 403), and the sense that methods must be equally flexible if they are to attune us to the complexity of contemporary religiosities and enact the religio-social worlds that we would like to see around us.

**Funding:** This research received no external funding.

**Institutional Review Board Statement:** Not applicable.

**Informed Consent Statement:** Not applicable.

**Data Availability Statement:** Not applicable.

**Conflicts of Interest:** The author declares no conflict of interest.

## Notes

[1]　Of note is the fact that these same methodological concerns—that some phenomena might be resistant to certain methodological procedures, and that research methods enact rather than represent states of affairs in the world—were crystallised in anthropology in James Clifford and George Marcus's collection of essays *Writing Culture* (1986). Stephen A. Tyler's essay 'Post-Modern Ethnography: From Document of the Occult to Occult Document' is exemplary of this impulse: "A post-modern ethnography is a cooperatively evolved text consisting of fragments of discourse intended to evoke in the minds of both reader and writer an emergent fantasy of a possible world of commonsense reality, and thus to provoke an aesthetic integration that will have a therapeutic effect. It is, in a word, poetry—not in its textual form, but in its return to the original context and function of poetry, which, by means of its performative break with speech, evoked memories of the *ethos* of the community and thereby provoked hearers to act ethically. Post-modern ethnography attempts to recreate textually this spiral of poetic and ritual performance. Like them, it defamiliarises common-sense reality in a bracketed context of performance, evokes a fantasy whole abducted from fragments, and then returns participants to the world of common sense—transformed, renewed, and sacralised. It has the allegorical import, though not the narrative form, of a vision quest or religious parable. The break with everyday reality is a journey apart into strange lands with occult practices—into the heart of darkness—where fragments of the fantastic whirl about in the vortex of the quester's disoriented consciousness, until, arrived at the maelstrom's centre, he loses consciousness at the very moment of the miraculous, restorative vision, and then, unconscious, is cast up onto the familiar, but forever transformed, shores of the commonplace world. Post-modern ethnography is not a new departure, not another rupture in the form of discourse of the sort we have come to expect as the norm of modernist esthetics' scientist emphasis on experimental novelty, but a self-conscious return to an earlier and more powerful notion of the ethical character of all discourse, as captured in the ancient significance of the family of terms "ethos", "ethnos" and "ethics". Because post-modern ethnography privileges "discourse" over "text", it foregrounds dialogue as opposed to monologue, and emphasizes the cooperative and collaborative nature of the ethnographic situation in contrast to the ideology of the transcendental observer" (Tyler 1986, pp. 125–26).

[2]　Interestingly, the move to posit the biological as the deep, sovereign truth that lies behind religion and culture in the way that many cognitivists do, suggests quite a narrow and reductive view of biology. According to Meloni, "if the social sciences are ill, biology looks like the therapy; if sociological investigations are thin and fragmented, biological knowledge is solid and cohesive", and if the social seems to be "an erratic, ephemeral entity, lacking firmer ground, what is required is to anchor it onto the firmer basis of evolutionary thinking and neurobiological facts" (Meloni 2014a, p. 733). Importantly, what Meloni calls

"post-genomic" biology challenges the view of the "biological as what is 'genetic', 'innate', 'prior to social', 'essential', 'universal', and 'invariable'" (Meloni 2014a, p. 732) towards the idea of the biological as "just another interactant" (Meloni 2014a, p. 742). In this new, post-genomic horizon, the social and the biological turn out to be increasingly porous, such that "there is no longer biology and culture but *hybrid resources* (interactants) in a unified developmental system" (Meloni 2014b, p. 606 italics in original).

3    However, like all approaches, discourse analysis also has certain effects and consequences which Foucault understood, and which led him to abandon the archaeology and turn to Nietzschean genealogy instead: "The task of the archaeologist is to describe in theoretical terms the rules governing discursive practices . . . the archaeologist claims to operate on a level that is free of the influences of both the theories and practices he studies . . . [but] the archaeologist's claim that he is totally detached . . . [is] problematic. Foucault's account of his own position with regard to the human sciences . . . undergoes a radical transformation. The investigator is no longer the detached spectator of mute discourse monuments. Foucault realizes and thematizes the fact that he himself—like any other investigator—is involved in, and to a large extent produced by, the social practices he is studying" (Dreyfus and Rabinow 1983, pp. 102–3). One of the effects of discourse analysis, then, is to generate a certain problematic objectivity whereby analysis takes place in a rarefied and unaccountable realm, beyond or outside the religious and social world that is being explored. A second effect is the tendency of discourse analysis to freeze the flows and eddies of religious and social worlds into neat, solid, manageable objects which can then be isolated for analytical scrutiny. For example, Cotter (2020), in his study of non-religion in Edinburgh's Southside, seems to have turned to discourse analysis specifically to confer upon non-religion the solidity and objectivity that its situational, relational and performative particularities deny. The turn to discourse, then, salvages a certain sense of religio-social worlds as stable and fixed while guaranteeing the objectivity and authority of intellectual labour.

4    Law and Urry (2004, p. 400) quote Heisenberg who suggested that "[w]hat we observe is not nature itself, but nature exposed to our method of questioning". Rovelli's (2021) brief history of quantum theory begins with Heisenberg's trip to the isolated Helgoland and makes the case for a radically relational and entangled universe, in which things are defined not in terms of some intrinsic substance or essence but rather in terms of the relations and practices through which they become available to us.

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
