# Peer review of "Transreligiosity and the Messiness of Religious and Social Worlds: Towards a Deleuzian Methodological Imagination for Religious Studies"

_religions, doi:10.3390/rel14040527_

Round 1
Reviewer 1 Report
Although I realize that the text is devoted to an issue of a methodological nature, I think it would be interesting for the reader to see the proposed method in action. Using the example of a selected issue, the author could show what the proposed method brings.
Author Response
Thanks for the comments/peer review. I accept all the points.
(a) I have adjusted the Introduction so that it introduces the article and doesn't simply repeat the Abstract.
(b) I have expanded the section drawing on my own research to clarify the argument.
(c) I have added appropriate references to crisis in the Introduction, the Conclusion and at a couple of points in the article.
Reviewer 2 Report
This is an interesting article. It emphasizes the postmodern and hyper-individualistic understanding of human nature applied to religious studies. More time needs devoted to developing how a modern understanding of religious studies empirical research methods reflects an imperialistic or monolithic understanding of religion while Panagiotopoulos and Roussou’s concept incorporates a more accurate or nuanced understanding one people's lived experiences regarding religion. Moving from the dominance of a cognitive science of religion to phenomenology indicates a move from cognitive determinism to affective or feeling driving the experience and understanding of religion. Developing these ideas and connecting them to the scientific practices of religious studies would be helpful for improving this paper.
Author Response
I have taken steps to improve the clarity of the article by revising the Introduction and Conclusion, by making clearer references to crisis and by expanding the discussion of my own fieldwork to improve the discussion of nomadology/Deleuze.
Reviewer 3 Report
This article presents an interesting and innovative approach to religious studies through the analysis of the concept of transreligiosity, proposed by Panagiotopoulos and Rous, as a means to rethink the classical and contemporary methodological debates surrounding this "discipline" as well as the methods of reflection derived from social practices. Through transreligiosity, the author manages to underline the relational-historical dimension of the religious as opposed to its merely substantialist consideration. He also shows very clearly how current research methods, founded on disciplinary inspirational approaches, experience great difficulties in describing social realities- like religions- that are complex, diffuse and messy. In this way, procedural approaches in order to relating concepts or religious research should be avoided, as the article states. Moreover, methods in religious studies need to be sensitive to the phenomenon of complexity, abandoning disciplinary perspectives that presuppose rules and protocols to be applied to an apparently pre-existing reality. The paper show this with extraordinary clarity.
Likewise, the author makes tremendously interesting explorations of the various contemporary methodologies (Phenomenology, hermeneutics, cognitive theory and discourse analysis) of access to the religious phenomenon, highlighting among them, for its clarity, the phenomenological approach proposed by Mircea Eliade.
However, the article presents some structural problems that the author should correct:
1. I consider that the abstract should somehow include a reference to nomad science, since it constitutes a section of its own in the article. On the other hand, it would be good if it were a little longer.
2. The introduction is a literal copy of the abstract, except for the final quote. From my point of view, the introduction should reflect and develop more extensively the context of the problem to be addressed in the paper. This is the mean weakness of the article.
3. The relationship between the nomad sciences section and the other sections is not sufficiently justified.
4. The conclusion could be more extensive and detailed.
Author Response
Thank you for taking time to review the article. I have revised the Introduction and Conclusion so that they better reflect the article and do not repeat the Abstract (and include the term 'nomadology'). I have also revised the section that draws on my own fieldwork to give the overall argument greater clarity.
Round 2
Reviewer 2 Report
This is a much improved paper. I am satisfied the authors addressed the revisions.
Reviewer 3 Report
The changes have improved the quality and clarity of the article. As a suggestion, I would only insist on the need to expand the abstract.